# Feasibility, Process, and Effects of Short-Term Calorie Reduction in Cancer Patients Receiving Chemotherapy: An Integrative Review

**DOI:** 10.3390/nu12092823

**Published:** 2020-09-15

**Authors:** Chia-Chun Tang, Hsi Chen, Tai-Chung Huang, Wei-Wen Wu, Jing-Mei Lin, Feng-Ming Tien

**Affiliations:** 1School of Nursing, College of Medicine, National Taiwan University, Taipei 10617, Taiwan; hchsichen@ntu.edu.tw (H.C.); weiwen@ntu.edu.tw (W.-W.W.); 2Division of Hematology, Department of Internal Medicine, National Taiwan University Hospital, Taipei 10617, Taiwan; tch01@ntu.edu.tw (T.-C.H.); b92401007@ntu.edu.tw (F.-M.T.); 3Department of Dietetics, National Taiwan University Hospital, Taipei 10617, Taiwan; kyomilin@ntuh.gov.tw

**Keywords:** integrative review, short-term calorie reduction, fasting, cancer, chemotherapy, calorie restriction

## Abstract

Recent preclinical studies have shown the potential benefits of short-term calorie reduction (SCR) on cancer treatment. In this integrative review, we aimed to identify and synthesize current evidence regarding the feasibility, process, and effects of SCR in cancer patients receiving chemotherapy. PubMed, Cumulative Index to Nursing and Allied Health Literature, Ovid Medline, PsychINFO, and Embase were searched for original research articles using various combinations of Medical Subject Heading terms. Among the 311 articles identified, seven studies met the inclusion criteria. The majority of the reviewed studies were small randomized controlled trials or cohort study with fair quality. The results suggest that SCR is safe and feasible. SCR is typically arranged around the chemotherapy, with the duration ranging from 24 to 96 h. Most studies examined the protective effects of SCR on normal cells during chemotherapy. The evidence supports that SCR had the potential to enhance both the physical and psychological wellbeing of patients during chemotherapy. SCR is a cost-effective intervention with great potential. Future well-controlled studies with sufficient sample sizes are needed to examine the full and long-term effects of SCR and its mechanism of action.

## 1. Introduction

Emerging evidence has shown that glucose and caloric intake have powerful impacts on health, in both the general and the critically ill population, including cancer patients [1,2,3,4]. High glucose levels can contribute to a vicious circle that affects cancer formation, treatment, and progression [5,6]. Recent expert opinions suggest that glucose reduction and calorie control could enhance cancer treatments and improve patient outcomes [7,8]. There are at least four proposed mechanisms of how calorie restriction (CR), or fasting, affects tumor growth and treatment effectiveness. First, CR increases tumor cells’ sensitivity to anticancer therapy by promoting apoptosis within tumors, which reduces levels of growth factors such as insulin-like growth factor-1 (IGF-1), and by inducing autophagy via the activation of AMP-activated protein (AMPK)/the mammalian target of rapamycin (mTOR) pathway. Second, in contrast, CR selectively protects normal cells from stress and toxicity of anticancer therapy because they react oppositely to the aforementioned interferences. Moreover, CR-induced autophagy may promote tissue regeneration. Third, by decreasing inflammation and increasing circulating T cells, CR establishes an environment that is unfavorable to tumor growth. Fourth, CR inhibits tumor growth by reducing the expression of factors that promote neovascularization of tumors [8,9,10,11,12,13].

Compared with a chronic 20–40% CR, which requires weeks to months to detect its effects on cancer progression, a short-term CR (SCR; for example, a calorie reduction of over 50% lasting no longer than a week) has shown immediate effects on enhancing the therapeutic effects of chemotherapy and protecting normal cells from drug toxicity [12,14,15]. SCR also seems to be safe, and does not cause weight loss, which is the main side effect of chronic CR [10]. Several in vivo (mouse models) and in vitro studies have demonstrated positive effects of SCR on suppressing tumor growth (for example, in pancreatic cancer and hepatocellular cancer) and enhancing the effects of chemotherapeutic agents (such as, doxorubicin, gemcitabine, and sorafenib). The in vivo studies have shown that SCR significantly increases chemotherapy effects by inhibiting tumor growth, cellular proliferation, and metabolism [12,14,15]. D’Aronzo and colleagues even demonstrated that SCR alone is just as effective as SCR plus gemicitabine in inhibiting pancreatic cancer cell migration in vitro and using animal models [12]. Some evidence has indicated that undertaking SCR (fasting for 24–72 h with access to water or eating a diet that mimics fasting) prior to chemotherapy protects normal cells, regulates glycemia, and enhances the therapeutic effects of chemotherapy [12,15,16]. Di Biase and colleagues found that SCR decreased doxorubicin-induced cardiotoxicity and prevented hyperglycemia in mice, thereby providing protection from glucose- and dexamethasone-dependent sensitization to doxorubicin [16].

Although the results from animal studies are promising and human trials have begun, clinical oncologists to date only provide universal and generic dietary guidelines to all cancer patients [17]. For example, in the latest nutrition guide published by the American Institute for Cancer Research, Livestrong Foundation, and Savor Health [18], the main nutrition recommendation for all cancer patients under treatment is to eat a healthy and clean diet. The European Society for Clinical Nutrition and Metabolism (ESPEN) guideline for patients undergoing drug treatment is to “ensure adequate nutritional intake” [19]. Several experts have pointed out that the level of evidence for these recommendations is low [17,19]. In fact, to our knowledge, no nutrition guidelines or recommendations have ever mentioned any form of SCR. This may be due to the early stage of clinical studies and the lack of systematic reviews that evaluate and synthesize current SCR evidence. The vague recommendation is insufficient to answer the necessary but unanswered question of “how to eat right?” In a survey (*n* = 1335), more than two thirds of the patients with cancer indicated that they had questions regarding nutrition or food intake [20]. In contrast, a considerable number of cancer patients (39–76%) have reported unmet needs regarding nutrition-related information or issues [21,22]. Therefore, the aim of this review is to identify and synthesize current evidence regarding the feasibility, process, and effects of SCR in cancer patients receiving chemotherapy. The findings from this review will identify areas for future research, aid in reexamining nutrition guidelines and enhance evidence-based clinical practice.

## 2. Materials and Methods

It is important to analyze all the available data for new concepts and underexplored research areas such as SCR. Therefore, the method of integrative review was selected; this allowed us to include as much evidence as possible, regardless of the study design and type of data. We followed the well-established review process described by Whittemore and Knafl, which included the following: problem identification, literature search, data evaluation and analysis, and presentation of the results [23].

### 2.1. Literature Search

We searched the following five databases for articles describing SCR in cancer patients undergoing chemotherapy: PubMed, CINAHL, Ovid Medline, PsychINFO, and Embase. Several combinations of Medical Subject Heading (MeSH) terms were used in different databases (Table 1). The original studies exploring the effects of SCR on cancer patients receiving chemotherapy were included only if they were written in English, included human cancer patients, and were peer-reviewed. We did not set any limits on the dates of publication and the final date of the search is the 6 August 2020. Articles were excluded if they did not meet any one of the aforementioned criteria or if they focused on the effects of food on drug pharmacokinetics. The eligibility of the literature was determined by screening the titles, then the abstracts, finally, a full-text review. In addition, the reference lists of each included article and the website ClinicalTrials.gov were searched to identify relevant studies. EndNote X8 was then used to sort citations and remove duplicates.

### 2.2. Data Evaluation and Analysis

We fully reviewed and rated the included literature in terms of its level of evidence and level of quality presented, which reflects the generalizability of a study. The definition of each level of evidence are presented in Table 2 which was modified from Wright and colleagues [24]. There are four level of research quality: good (the risk of bias is very low and the results are considered to be valid), fair (the study is susceptible to some bias deemed not sufficient to invalidate its results), poor (there is a significant risk of bias), and not to be analyzed (there is a fatal flaw) [25,26]. Because the designs of the included studies vary, we employed four scales to evaluate the quality of the studies. Quantitative studies were evaluated on the basis of Quality Assessment Tools developed by methodologists from the NHLBI and Research Triangle Institute International. Specifically, the Quality Assessment of Controlled Intervention Studies [27] was used to evaluate randomized controlled trials, the Quality Assessment Tool for Observational Cohort and Cross-Sectional Studies [28] was chosen to assess prospective cohort studies, and the Quality Assessment Tool for Case Series Studies [27] was selected for case studies. Instructions for Evaluating Qualitative Literature [26] were employed for qualitative studies. Studies that met 75–100% criteria were determined to be of good quality while 50–74% criteria met signified a fair quality and 25–49% criteria met indicated poor quality. Next, study information was collected and categorized in a data collection file prepared by C.T. using Microsoft Word. Specifically, three kinds of information were collected: study characteristics (design, population, fasting plan, and type of chemotherapy), type of outcome measurements, and main study outcomes. All research activities were independently performed by C.T. and H.C. In case of discordant opinions, the research team discussed and solved these issues in regular meetings.

## 3. Results

Initially, 311 articles were identified. After removing duplicates (*n* = 3), 308 articles were screened by title, which resulted in a total of 67 articles for abstract screening. Using the established criteria, 60 articles were excluded. Among the 60 articles, 60% (*n* = 36) were not complete original research articles; 33% (*n* = 20) presented irrelevant content; 5% (*n* = 3) did not include human samples; and 2% (*n* = 1) were not written in English. The remaining seven studies that were retained for full-text review were all included in the analysis (Figure 1).

### 3.1. Study Characteristics

Among the seven studies included in this review (Table 3), one is a qualitative study and others have a quantitative design, including a case study, a cohort study, and randomized controlled trials (RCT, n = 4). The sample sizes ranged from 13 to 129. Five studies focused on gynecologic cancer populations [29,30,31,32,33] and the other two involved various types of cancer. In the five studies that stipulated strict timelines for SCR, the total period ranged between 24 and 96 h, with SCR typically starting 24–72 h before the chemotherapy and lasting for about 24 h after the completion of chemotherapy [29,30,32,33,34]. The other two studies observed participants’ self-determined reduction practices, and thus presented large variations in the SCR timeframe—the patients started SCR 24–140 h prior to chemotherapy and ended it 5–56 h following chemotherapy [31,35]. The actual number of calories consumed during the practice of SCR differed across studies. Most studies required the participants to fast, allowing only non-caloric beverages. One study offered a rescue option to consume less than 200 kcal a day if fasting symptoms became apparent [34]. Bauersfeld et al. set the daily maximum total intake at 350 kcal [29] and de Groot et al. [32] designed a fasting mimicking diet with decreasing calorie amount over three days (200–1200 kcal). On the other hand, Zorn et al., instructed a group of patients to consume a 6-day normocaloric ketogenic diet before water fast. While a case study mentioned that some of their participants ate nothing except for water and vitamins [35], other studies did not specify if any nutritional supplements were used. The participants received various types of chemotherapy drugs and regimens, including taxanes, platinum, alkylating, anthracycline, antimetabolites, and IgG1 antibody. In terms of the level of evidence of the quantitative studies, the majority was level II small RCTs or cohort study (*n* = 4); others were level I RCT (*n* = 1) and level IV case series (*n* = 1) [24]. Using the aforementioned quality scales to evaluate, more than half of the studies had fair or poor quality (Table 4 and Table 5). Only two studies were of good quality, including one RCT and one qualitative study (data not shown in table) [31,32]. The most obvious threats to the quality of RCT studies were the high drop-out rates and low adherence.

### 3.2. Outcome Measurements

The following two categories of SCR outcomes were evaluated: safety/tolerance and overall effect. Specifically, the safety and tolerance of SCR were measured on the basis of the reasons for non-compliance with SCR, symptoms that were directly induced by SCR, and the change in nutrition or metabolism status. The effects of SCR were evaluated on the basis of its protective or regenerative effect on normal cells, ameliorative effect on inflammation, and sensitizing effect on tumor cells. The protective or regenerative effect on normal cells were evaluated on the basis of disease- or chemotherapy-associated side effects, quality of life, DNA damage in healthy cells, and hematological function. The reduction in inflammation was measured on the basis of the inflammatory response. The sensitizing effect on tumor cells to chemotherapy was evaluated using endocrine parameters and treatment outcomes. In addition to blood samples, several tools, such as Common Terminology Criteria for Adverse Events (CTCAE), the Functional Assessment of Chronic Illness Therapy-General (FACIT-G), The European Organization for Research and Treatment of Cancer quality of life questionnaires (EORTC QLQ), and the Functional Assessment of Chronic Illness Therapy- Fatigue (FACIT-F) were employed to assess the side effects, symptoms, and quality of life. The researchers followed these variables across multiple cycles of chemotherapy in the following periods: “before each SCR and/or chemotherapy”, “hours to days after each chemotherapy”, “about a week after each chemotherapy”, “at the end of chemotherapy treatment”, and “6-month after treatment”.

#### 3.2.1. Safety and Tolerance of SCR

All studies concluded that SCR was safe, well-tolerated, and feasible [29,30,31,32,33,34,35]. More importantly, many participants expressed a strong motivation to undertake SCR and a desire to continue the practice in the future because of the perceived benefits of SCR, which included an increased sense of control [29,30,31].

The reported success rate of completing one cycle of SCR was above 80% [29,30]. However, the adherence decreased to below 50% when the researchers followed for more than three cycles [32,33]. Excluding non-SCR related symptoms (such as recurrent febrile neutropenia) and personal factors (such as forgetting, changing chemotherapy plan, and others), the reasons for withdrawal included headache, hyperventilation, weakness, failure to regain weight, aversion to fasting nutrition, and social constraints [29,30,32,33,34,35]. The qualitative study also reported social constraints as barriers to SCR—the patients who performed self-initiated SCR indicated that the protocol interfered with meal-sharing in their social lives. They also highlighted that the uncertainty surrounding the effects of fasting could be a barrier to SCR. In contrast, anxiety regarding hospitalization and positive social support might facilitate fasting behavior [31].

All researchers concluded that the possible side effects of SCR were mild, and that they either did not interfere with daily activities or did not require special treatment. The following side effects were noted: hunger, fatigue, dizziness, headache, hypoglycemia, weight loss, hyponatremia, orthostatic reaction or hypotension, and nausea after taking broth or juice [29,33,34,35]. Although weight loss may be an expected side effect of SCR, the studies showed that the loss of body weight was absent or minimal (about 6–7 pounds, <5%) [29,33,35], and that it was regained quickly after resuming a normal diet [29,35]. While pilot studies reported that no obvious changes in parameters related to nutrition and metabolism, such as prealbumin, insulin, and glucose, were observed [30,34], larger RCTs indicated that glucose and insulin were significantly lower in SCR groups before and during the treatment than controls [32,33]. The duration of fasting significantly affected ketone levels: de Groot and colleagues noted a decreasing trend in β-hydroxybutyrate levels (a type of ketone body) in 24-h fasting groups and an increasing trend in groups that fasted for more than 48 h [30]. The same research group later reported that ketone bodies were more likely to be positive in patients that performed SCR compared to regular diet [32]. In one study that examined body composition, the results showed decreased bioelectrical impedance analysis (BIA) fat mass, BIA body cell mass, mean BIA phase angle, and increased BIA extracellular cell mass [33].

#### 3.2.2. Effects of SCR

The results of the six quantitative studies show mixed but overall positive findings regarding the effects of SCR. Most of the studies focused on SCR’s protective or regenerative effect on normal cells, including chemotherapy-related side effects or symptoms [29,30,32,33,34,35], quality of life [29,30,31,32,33,35], hematological function [30,33,34,35], and DNA damage [30,32,34]. Five studies also examined endocrine parameters and/or treatment outcomes [30,32,33,34,35] to evaluate the sensitizing effects of tumor cells to chemotherapy. De Groot and colleagues measured inflammatory response [30].

##### SCR’s Protective or Regenerative Effect on Normal Cells

Several studies suggested that SCR significantly reduces multiple chemotherapy-related side effects, such as nausea, vomiting, stomatitis, fatigue, headache, and overall symptom burden [29,33,34,35], and improved quality of life [29,35]. However, some did not find a significant reduction in side effects [30,32] or an improvement in the quality of life [30,32,33]. Zorn et al. pointed out a significant relationship between SCR and fewer chemotherapy postpones. The findings from a qualitative study that examined patients’ motivation of self-initiated SCR reported that patients started SCR because they thought that it could mitigate the side effects of chemotherapy [31]. In fact, most of the patients reported positive physiological effects after fasting, and half of them experienced psychological benefits such as a reduction in feelings of uncertainty and anxiety [31].

To determine how SCR preserves or regenerates hematological function, the number and changes of erythrocytes, thrombocytes, and leukocytes were examined. All the studies that examined hematological function reported the protective effect of SCR [30,33,34,35], although the result from one study was insignificant [34]. Specifically, one study found that the erythrocyte and thrombocyte counts were significantly higher in the SCR group than in the control group one week or even 21 days after chemotherapy [30]. The results from another study showed a significantly milder neutropenia in patients who had fasted for longer than 48 h than in patients who had fasted for 24 h [34]. Zorn et al. (2020) found a significant decrease in mean corpuscular cell volume (MCV) and mean corpuscular hemoglobin (MCH).

Three studies looked at SCR’s protective effect on chemotherapy-induced DNA damage, which was based on peripheral blood mononuclear cells. The results are encouraging [30,32,34]. Specifically, while DNA damage was obvious in all patients immediately after chemotherapy, patients who had fasted showed less chemotherapy-induced DNA damage 30 min to seven days later compared to the non-fasting group [30,32]. Further, one study that compared outcomes of 24-, 48-, and 72-h fasting specified that this protective effect was only observed in participants who had fasted for 48 h or longer [34].

##### Sensitizing Tumor Cells to Chemotherapy

As IGF-1, insulin-like growth factor-binding protein (IGFBPs), thyroid-stimulating hormone (TSH), triiodothyronine (fT3), and free thyroxine (fT4) were evaluated, a trend of decreasing IGF-1 [30,32,33,34], decreased fT3, and increased fT4 was found [33]. These indicators were measured at baseline, after fasting (but before chemotherapy) [30,34], and 24 h after chemotherapy [34]. In terms of pathological responses, the results from one study that involved a small group of patients showed no obvious impact of SCR on chemotherapy [34]. However, a large RCT showed that three times more partial or complete pathological responses were observed in patients performing SCR than in patients eating a regular diet [32]. From patients’ perspectives, they indicated that they performed SCR because it could improve chemotherapy efficacy [31]. SCR did not have a significant effect on other parameters, such as inflammatory response [30,33].

## 4. Discussion

Taken together, the results indicate that SCR during chemotherapy is not likely to cause significant adverse effects, and is possible to alleviate treatment-induced side effects, improve quality of life, and stabilize hematological responses. Based on these results, SCRs are worth consideration for larger human trials; however, more high-quality RCTs are necessary before making relevant clinical practice recommendation.

The first important and clear takeaway is that SCR is feasible and well tolerated in cancer patients undergoing chemotherapy, in accordance with researchers who advocate for SCR [8,10]. The side effects directly caused by SCR were rare, and (if any) mild. Though weight loss and malnutrition may be the most worrisome side effects of SCR, the studies show that weight loss is minimal and reversible, and most nutrition parameters (such as prealbumin) remained stable during and after SCR [34]. Despite the minor side effects, the studies’ participant retention rates remains a big challenge. In addition, SCR has not yet been thoroughly examined in various types of cancer, male patient groups, and ethnically diverse patient populations. Ethnic or cultural factors play an important role in performing SCR, as eating behavior is closely associated with cultural beliefs [36]. Indeed, some of the reviewed studies showed that one of the barriers to continuing SCR is the social constraint when eating with others, since eating can be considered as a social activity and not only as a means to meet nutritional needs [31,34]. Since only one reviewed study addressed bioelectrical impedance analysis [33], future research may need to consider monitoring nutritional status more aggressively, such as by measuring the change in lean body mass [37].

Corresponding to Lee and Longo’s definition of SCR [8,10], the studies that set an SCR regimen required the participants to stay below 50% of the recommended daily calorie intake for no more than a week. Apart from this rough recommendation, it is necessary to discuss whether there is a more precise and appropriate amount and duration of calorie reduction. In the reviewed studies, protocols of reducing calorie intake to zero or providing a 200–400 kcal calorie intake were achievable. However, when compared with a zero-calorie intake, providing a small calorie intake or fasting-mimicking diet caused additional adverse effects, such as aversion or nausea to the provided nutrition [29,38]. On the other hand, the patients showed strong motivation for fasting and indicated that the anxiety of hospitalization automatically lowered their interest in eating [29,30,31]. Thus, it seems that shortly reducing the calorie intake to nearly zero during chemotherapy can be physically and psychologically acceptable to cancer patients. Future studies are needed to compare the pros and cons of water fasting, low calorie intake (<350 kcal) and fasting-mimicking diet. Although it is outside the scope of this review, comparing the outcomes of different kinds of diet modification is an important future work. For example, it seems that a high-fat, moderate-to-low-protein, and very low-carbohydrate ketogenic diet is effective against cancer [39,40]. Researchers have also proposed that fructose, amino acid, methionine, or serine restriction may have impacts on cancer treatment [4]. With regard to the SCR duration, though all the studies arranged the SCR around chemotherapy, one study that compared 24-, 48-, and 72-h fasting periods showed that groups that fasted for more than 48 h had the least DNA damage in healthy cells [34]. This result is similar to previous findings that show that fasting for longer than 72 h followed by refeeding can protect hematopoietic stem cells from the chemotherapy-induced toxicity and stimulate the proliferation and rejuvenation of old hematopoietic stem cells [41]. More work comparing the effects of different SCR durations are needed.

Our findings show clues regarding one of the aforementioned mechanisms [8,13]—the way SCR selectively protects normal cells from the stress and toxicity of anticancer. Most of the reviewed studies showed that undertaking SCR with chemotherapy, even for as short a period as a few days, could have a protective effect of healthy cells, which results in improving the overall quality of life and alleviating drug-induced side effects, including physical symptoms, nadir, and DNA damage to normal cells [29,30,33,34,35]. A couple of studies tried to find the association between SCR and tumor cells’ sensitivity to anticancer therapy [32,33,34,35]. The researchers measured IGF-1 or observed pathological response and imaging reports. Although a decreasing trend in IGF-1 level and a better pathological response were reported [32,33,34,35], the researchers did not arrive at a definite conclusion due to the limited number of studies and sample size. Then again, only one of the reviewed studies measured the inflammatory response, and it found no significant change [30]. Thus, it is difficult to conclude whether SCR had the potential to sensitize tumor cells to chemotherapy or facilitate the establishment of an environment against tumor growth. Moreover, while one of the proposed mechanisms of action is related to SCR-induced autophagy, it is imperative to notice the possible two-sided effects of autophagy modulation in tumor cell. Autophagy has the potential to promote tissue regeneration in both normal and tumor cells which may limit the effectiveness of chemotherapy [42]. More studies are needed to (1) explore the mechanism of action, (2) observe biological indicators 48 h or more after fasting, and (3) ensure a sufficient sample size. In addition, using a method that is sensitive to glucose metabolism, such as an FDG-PET/CT scan, may capture the treatment effects more precisely.

A new benefit of SCR has emerged from the results of the qualitative study: SCR improves patients’ psychological well-being by empowering them to restore self-control, be proactive, and feel less uncertain and anxious [31]. The positive psychological impacts of fasting have also been observed in healthy women, who experienced an increased sense of achievement, reward, pride, and control [43]. Psychological benefits should be important considerations for future clinical practice and research.

The inherent limitation of this review is the small and narrow study sample. As SCR during chemotherapy is a developing concept, human research has been conducted within the past ten years, and only in certain population (mostly female breast cancer) and geographical areas (U.S.A., Germany, the Netherlands, and France). The generalizability of the results is further precluded because cancer patients with nutritional issues or in a poor condition were automatically excluded from the studies. Because SCR had to be performed with chemotherapy, longer chemotherapy regimens could not be examined.

## 5. Conclusions

While growing evidence has shown hopeful effects of SCR in in vivo experiments and cancer patients, this study is the first to synthesize current evidence on SCR performance during chemotherapy in humans. Our findings suggest that the harm is manageable and that the benefits are worth investigating. While some RCTs are ongoing [38,44,45], more well-controlled studies with diverse ethnicities and cancer types are needed to confirm the effects of SCR and to refresh nutrition guidelines. A long-term follow-up would provide useful information regarding treatment effects and long-term side effects, yet the researchers need to overcome several challenges, including the low compliance rate. SCR should be an important consideration in the future, as it is cost-effective and potentially linked to many clinical outcomes. For example, SCR may be a solution for managing chemotherapy-related toxicity or hyperglycemia [8,46]. Clinicians’ close follow-up on the emerging evidence of SCR would provide perspectives for their current practice.

## Figures and Tables

**Figure 1 nutrients-12-02823-f001:**
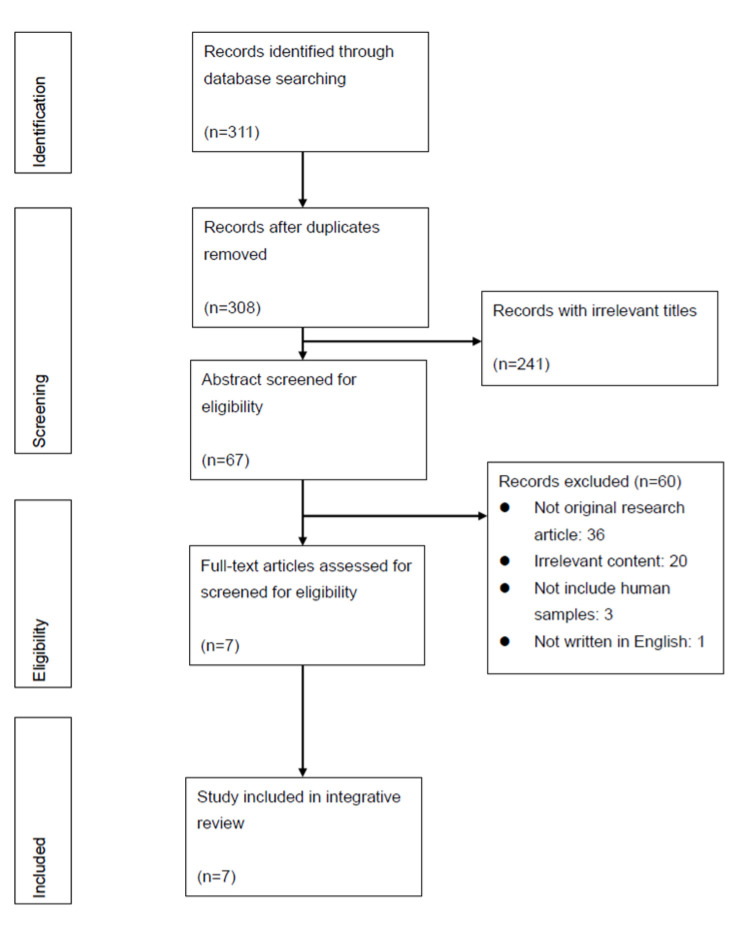
PRISMA Diagram of Search Results and Screening Process.

**Table 1 nutrients-12-02823-t001:** Searched databases, searching strategies, and the number of initial results.

Databases	Searching Strategies: Combination of Medical Subheadings	Initial Results
PubMed	(“fasting” OR “calorie restricted”) AND “chemotherapy”	238
Ovid Medline	(“fasting” OR “diet, carbohydrate-restricted” OR “calorie restriction”) AND (“maintenance chemotherapy” OR “induction chemotherapy” OR “consolidation chemotherapy” OR “chemotherapy, adjuvant” OR “chemotherapy, cancer, regional perfusion”)	9
CINAHL	(“fasting” OR (“preprocedural Fasting” OR “restricted diet” OR “diet, reducing” OR “diet, low carbohydrate”) AND (“chemotherapy, cancer” OR “chemotherapy, adjuvant” OR “chemotherapy care (Saba CCC)” OR “chemotherapy management (Iowa NIC)” OR “antineoplastic agents, combined”)	7
PsychINFO	(“calories” OR “dietary restraint”) AND “chemotherapy”	38
Embase	“caloric restriction” AND “cancer chemotherapy”	19

**Table 2 nutrients-12-02823-t002:** Level of Evidence.

Level	Definition
I	Randomized controlled trial
II	Prospective cohort study or Poor-quality randomized controlled trial
III	Case-control study or Retrospective cohort study
IV	Case series
V	Expert opinion

*Note.* Modified from Wright, Swiontkowski, and Heckman (2003) [24].

**Table 3 nutrients-12-02823-t003:** Information of Reviewed Articles: Type of Design, Method, and Maine Results.

First Author, Year (Country)	Goal: Research Design	Sample Size, Population, Exclusion Criteria	Calorie Reduction Plan	Chemotherapy Regimen	Measuring Time and Outcome Measurements	Main Results
Dorff, 2016 (USA)	Determine the safety/feasibility of fasting prior to C/T: Cohort study (24/48/72 h.)	- *n* = 20 (6–7 patient/cohort) - various cancer types/stages - Exclusion: DM, BM ≤ 20.5, recent BW loss > 10 kg	**Dose/time**: escalating fast, up to 72 h (24 h before C/T completion → 48 h before C/T completion → if safe/feasible, then continue with 72 h (48 h before and 24 h after); if not, then try 48 h with specific low-calorie diet (repeat for at least 2 C/T cycles)) **Content**: NPO except for water and non-caloric beverage and rescue (<200 kcal/24 h. if fasting symptoms present)	- ≤2 days of Platinum-based combination C/T without concurrent radiation - May have begun C/T but still have 2 or more cycles - Standard antiemetic	○ Measuring before C/T, after fast, and 24 h after C/T completion: - Nutrition and metabolism status: Prealbumin, Insulin, Glucose, Ketones (β-hydroxybutyrate) - Side effects and fasting-related toxicities (CTCAE v4.0) - Hematological function - Endocrine parameters: IGF-1, IGFBPs - DNA damage: peripheral blood mononuclear cells - Treatment outcome: pathologic responses	○ Safety and tolerance of SCR: safe and feasible - Reasons for non-compliance: forget, social constraints, change of C/T plan, fail to regain weight - Nutrition and metabolism status *: β-hydroxybutyrate decreased in the 24-h group while it increased in 48- and 72-h groups ○ Main effects of SCR: - Effects on side effects, symptoms, and QOL *: decreasing C/T-related toxicity (nausea and vomiting) in all groups - Effects on hematological function: Insignificant trend of decreasing C/T-related grade 3–4 neutropenia - Effects on endocrine parameters: decreased but not-significant trend of IGF-1 - Effects on DNA damage in healthy cells *: mitigated in subjects who fasted for ≥48 h. - Effects of the treatment: no effects
Safdie, 2009 (USA)	Examine the safety of fasting before and after chemotherapy: Case study	- *n* = 10 - Various cancer types/stages	(Vary by cases) **Does/time**: 48–140 h prior to and/or 5–56 h following C/T (self-selected C/T cycles) **Content**: Some NPO except for water and vitamin, others unspecified **Control**: self-control	Individualized	○ At unspecified time points: - Self-reported symptoms: fatigue, weakness, hair loss, headaches, nausea, vomiting, diarrhea, abdominal cramps, mouth sores, dry mouth, short-term memory impairment, numbness, tingling, neuropathy motor - Hematological function: WBC, ANC, platelets - Treatment outcomes: CT-PET scan (one case)	○ Safety and tolerance of SCR: well-tolerated - Side effects: slight dizziness, hunger, and headache which did not interfere with daily activities. - Nutrition and metabolism status: weight loss was about 6–7 pounds which were regained quickly after resuming normal diet ○ Main effects of SCR: - Effects on side effects, symptoms, and QOL: self-reported reduction in multiple chemotherapy-induced side effects - Effects on hematological function: better recovery of blood counts, including less severe or shorter nadir of WBC/ANC/platelets - Effects of the treatment: better response to C/T in one patient
Bauersfeld, 2018 (Germany)	Examine the feasibility and effects of QOL of short-term fasting during C/T: Randomized, individually controlled trial	- *n* = 34 - Breast/ovarian cancer - Exclusion: BMI < 19, WHO performance status > 2, life expectancy < 3 months, DM, MI, stroke or pulmonary embolism within 3 months, unstable heart disease, renal failure, eating disorder, dementia, psychosis, impaired physical mobility	**Dose/time**: 60 h (36 h before and 24 h after C/T) **Content**: Unrestricted amounts of water, herbal tea, 2 × 100 cL vegetable juice and small standardizes quantities of light vegetable broth with a maximum total energy intake of 350 kcal/day **Control group**: self-controlled (group A: fast for the first half of C/T cycles (2 or 3 cycles) followed by normal diet); group B: vice versa sequence)	- 4–6 cycles of C/T: Taxanes, Platinum, Alkylating, Anthracycline, Antimetabolites, IgG1 antibody - Standard antiemetics and medication: dexamethasone and 5HT3 inhibitors	○ Baseline and 8 days after each C/T cycle: - Side effects and fasting-related toxicities: FACIT-G, FACIT-F ○ During and at the end of fasting: - Adverse events	○ Safety and tolerance of SCR: safe and well tolerated. - Reasons for non-compliance: headache, hyperventilation, weakness, aversion to fasting nutrition (n = 5, 10%) - Side effects: headache, hunger, nausea after intake of broth or juices, and orthostatic reaction; all were of low grade which did not interfere with daily activities - Nutrition and metabolism status: no significant changes in weight - More than 80% participants agreed that the fast was effective and wanted to continue the practice during C/T ○ Main effects of SCR: - Effects on side effects, symptoms, and QOL: less compromised QOL and reduced fatigue (Group A demonstrated a statistically significant and clinically meaningful benefits of fast on QOL and fatigue, while Group B only show clinically meaningful difference of the positive effect on QOL for fast intervention)
de Groot, 2015 (The Netherlands)	Identify the effects of 48 h fasting on C/T, including side effects, hematological parameters in breast cancer patient receiving TAC: Randomized controlled trial	- *n* = 13 - Stage II/III breast cancer - Exclusion: BMI < 19, WHO performance status > 2, life expectancy < 3 months, adequate function of bone marrow, liver, renal, and heart, DM	**Dose/time**: 48 h fasting (24 h before and after starting C/T) **Content**: NPO except for water or coffee/tea without sugar **Control group**: Eat according to the guidelines for healthy nutrition (n = 6, minimum of 2 pieces of fruit per day)	○ 6 cycles of (neo)-adjuvant TAC (docetaxel/doxorubicin/cyclophosphamide) ○ Anti-emetic agent and medication: dexamethasone, 5-HT3 receptor antagonist	○ Baseline (2 weeks before C/T), day 0 (prior to C/T) plus 30 min after C/T completion and day 7 of administration (only for hematological function, CRP, and DNA damage): - Nutrition and metabolic status: insulin, glucose - Hematological function: erythrocyte-, thrombocytes-, leukocyte count - DNA damage: γ-H2AX - Endocrine parameters: IGF-1, IGFBP3, TSH, triiodothyronine, free thyroxine - Inflammatory response: CRP ○ During C/T: self-reported side effects and CTCAE	○ Safety and tolerance of SCR: Participates were motivated to fast and the fast was well-tolerated and safe - Reasons of non-compliance: 2 withdraw at the 3rd cycle of C/T due to non-fasting-related signs (i.e., pyrosis and recurrent febrile neutropenia) - Nutrition and metabolism status: no significant changes ○ Main effects of SCR: - Effects on side effects, symptoms, and QOL: no significant effects - Effects on hematological function *: protect from C/T-related toxicity - Effects on endocrine parameters and inflammatory response: not significant - Effects on DNA damage in healthy cells (lymphocytes and myeloid cells) *: protect and promote recovery
de Groot, 2020 (The Netherlands)	Evaluate the impact of FMD on toxicity as well as on the radiological and pathological response to chemotherapy for breast cancer: Randomized controlled trial	- *n* = 129 - HER-2 (-), stage II/III breast cancer with WHO performance stage 0–2, BMI < 19 kg/m^2^ - Exclusion: DM, allergies to designed food content, function impairment (bone marrow reserve, liver, renal, cardiac)	**Dose/time**: 4-day plant-based low amino-acid substitution diet (FMD, 3 days prior to and on the day of C/T) **Content**: decreased calorie intake from FMD (1200 kcal at day 1, 100 kcal at day 2–4) **Control group**: regular diet	○ 8 cycles of (neo)-adjuvant (docetaxel/doxorubicin/cyclophosphamide) or 6 cycles of (neo)-adjuvant FEC-T (5-fluorouracil, epirubicin, cyclophosphamide) ○ Dexamethasone was given before C/T only for control group	○ Baseline (day-1 or 0 of each chemotherapy): - Nutrition and metabolic status: insulin, glucose, ketone - Endocrine parameters: IGF-1 ○ Baseline (day-1 or 0 of the first cycle of chemotherapy) & 3 h after start of C/T: DNA damage: γ-H2AX ○ Baseline, halfway, at the end of therapy, and 6-month follow-up: EORTC QLQ-C30, ○ During C/T: CTCAE v4.03 ○ Halfway and at the end of the therapy: pCR ○ Halfway, at the end of the therapy, and 6-month follow-up: distress thermometer	○ Safety and tolerance of SCR: FMD was well-tolerated and safe - Reasons of non-compliance: the compliance decreased along with the C/T cycles (81.5% to 20% from cycle 1 to 8). The main reason of non-compliance was dislike of distinct components of the diet - Side-effects: no differences in toxicity - Nutrition and metabolism status *: lower insulin and glucose, ketones in urine (+) ○ Main effects of SCR: - Effects on side effects, symptoms, and QOL: not significant - Effects on endocrine parameters * and inflammatory response: lower IGF-1 - Effects on DNA damage in healthy cells *: protect and promote recovery - Pathological response *: better pCR
Zorn, 2020 (German)	Evaluate the influence of 96-h fasting on chemotherapy-induced toxicities in patients with gynecological cancer: controlled cross-over trial	- *n* = 30 - Gynecological cancer - Exclusion: malnutrition, eating disorders, DM, gout, severe cardiovascular disease, pregnancy or lactation, parental nutrition, administration of steroids or IGF-1 receptor blockers	**Dose/time**: 96 h fasting (72 h before and 24 h after starting C/T) or 6-day normocaloric ketogenic diet plus 96 h fasting **Content**: 25% of daily calorie requirement (400–600 kcal/day) with macronutrients revealed to a ketogenic composition **Control group**: everyone served as their own controls (2–3 cycles of SCR and 2–3 cycles of normal diet)	○ Paclitaxel/carboplatin ○ Epirubicin/cyclophosphamide ○ Docetaxel/cyclophosphamide	○ Baseline (before C/T), at each C/T, 3 weeks after the final C/T cycle: - Side effects and fasting-related toxicities: self-reported, CTCAE v4.0, EORTC QLQ-C30, EORTC QLQ-CIPN20, FACIT-Fatigue v 4.0 - Nutrition and metabolic status: insulin, body composition - Hematological function: erythrocyte-, thrombocytes-, leukocyte count - Endocrine parameters: IGF-1, TSH, triiodothyronine, free thyroxine - Inflammatory response: CRP	○ Safety and tolerance of SCR: well-tolerated and safe - Reasons for non-compliance: 2 withdraw because of fating-related discomfort and 19 withdraw due to non-fasting-related reasons - Side-effects: hunger, dizziness, weakness, and headache were mild - Nutrition and metabolism status *: reduction in insulin, BIA fat mass, weight (<5%), mean BIA cell mass, mean BIA phase angle; increase in BIA extracellular cell mass ○ Main effects of SCR: - Effects on side effects, symptoms, and QOL *: decreased symptoms such as stomatitis, headache, weakness, and overall symptom severity - Effects on hematological function *: decreased MCV and MCH - Effects on endocrine parameters and inflammatory response *: reduction in IGF-1, triiodothyronine; increase in free thyroxine
Mas, 2019 (France)	Explore the motivations to fast among cancer patients: Qualitative study	- *n* = 16 - Breast cancer	**Dose/time**: Having performed at least one 24 h fast before C/T within a year (duration ranges from a day and half to 7 days; C/T cycles range from one to 10 months). **Content**: not specified	Not mentioned	- Qualitative description of reason to fast, modalities of the fast, experience of fasting, related social support, barriers and facilitators of fasting - Satisfaction	○ Safety and tolerance of SCR: Patients believed that fasting is an efficacious non-conventional medicine that helps to reduce side effects of C/T for breast cancer. Patients expressed high level of satisfaction toward fasting. ○ Main effects of SCR: - Six themes emerge: Main reason to fast: to lower the negative side effects of C/T, to regain control/act proactively during treatment (thus reduce the feelings of uncertainty and anxiety), improve C/T efficacy Alternative authorities to the oncologist: conventional health care professions and other cancer patients’ experience of fasting Adapting the fast to social and lifestyle constraints: fasts were always performed with C/T Fasting effects felt during chemotherapy: most of the patients reported positive physiological effects (especially nausea and vomiting) and about half experience psychological benefits Barriers to (uncertainty of the effect of fasting, interference of meal sharing social life) and facilitators (anxiety regarding hospitalization, positive social support) of fasting during C/T Seeking a more integrative medicine (although not supported by medical providers)

*Note.* 1. DM, Diabetes Mellitus; SCR, Shor-term Calorie Reduction; BMI, Body Mass Index; BW, Body Weight; kg, kilogram; kcal, kilocalorie; h, hour; n, size of sample; CTCAE, Common Terminology Criteria for Adverse Events; QoL, Quality Of Life; WBC, White Blood Cell; ANC, Absolute Neutrophil Counts; CT-PET, Computed Tomography-Positron Emission Tomography; MI, Myocardial Infarction; TAC, docetaxel/doxorubicin/cyclophosphamide; IGFBP3, Insulin Growth Factor Binding Protein 3; C/T, Chemotherapy; TSH, Thyroid-Stimulating Hormone; CRP, C-Reactive Protein; IGF-1, Insulin-like Growth Factor-1; IGFBPs, IGF Binding Proteins; NPO, Nothing by Mouth; cL, centilitre; FMD, fast mimic diet; pCR, pathological complete response; CTCAE, Common Terminology Criteria for Adverse Events; EORTC QLQ C30, The European Organization for Research and Treatment of Cancer quality of life questionnaire; EORTC QLQ CIPN20, The European Organization for Research and Treatment of Cancer quality of life Chemotherapy-Induced Peripheral Neuropathy; FACIT-F, the Functional Assessment of Chronic Illness Therapy-Fatigue; MCV, Mean corpuscular volume; MCH, Mean Corpuscular Hemoglobin. 2. * indicated that findings reached statistical or clinical significance.

**Table 4 nutrients-12-02823-t004:** Quality and Evidence Level of Cohort Study and Case Report.

Quality Rating Criteria for Cohort Study	Study	Quality Rating Criteria for Case Report	Study
	Dorff, 2016		Safdie, 2009
Research question/objective was clearly stated	Yes	Research question/objective was clearly stated	No
Study population was clearly specified/defined	Yes	Study population was clearly specified/defined	Yes
Participation rate of eligible persons was ≥50%	Unclear	Cases were consecutive	No
Prespecified Inclusion/exclusion criteria	Yes	Subjects were comparable	No
Justification of sample size/power/variance/effect size	No	Intervention was clearly described	Yes
Exposure(s) measured prior to outcome(s) evaluation	Yes	Clearly defined, valid and reliable outcome measures	No
Sufficient timeframe to see a possible association	Yes	Adequate length of follow-up	Yes
Examine different exposure levels as related to the outcome	Yes	Well-described statistical methods	Not applicable
Clearly defined, valid and reliable exposure measures	Yes	Well-described results	Yes
Assessed the exposure(s) more than once over time	Yes		
Clearly defined valid and reliable outcome measures	Yes		
Outcome assessors were blinded to the exposure status of participants	Unclear		
Loss to follow-up was 20% or less	Yes		
Key potential confounding variables measured and adjusted statistically	No		
Suggesting Quality (% of criteria met)	Fair (71%)	Suggesting Quality (% of criteria met)	Fair (50%)
Level of Evidence	II	Level of Evidence	IV

*Note.* Level of quality was defined as: Good Quality (75–100% criteria met), Fair (50–74% criteria met), Poor (25–49% criteria met).

**Table 5 nutrients-12-02823-t005:** Quality and Evidence Level of Randomized Controlled Trial.

Quality Rating Criteria	Studies			
	Bauersfeld, 2018	de Groot, 2015	de Groot, 2020	Zorn, 2020
Study was described as randomized or an RCT	Yes	Yes	Yes	No
Adequate randomization	Yes	Yes	Yes	No
Concealed treatment allocation	Yes	Unclear	Yes	Unclear
Study participants and providers were blinded to group assignment	Not applicable	Not applicable	Not applicable	Not applicable
People assessing the outcomes were blinded to the assignments	Unclear	Unclear	Yes	Unclear
Groups were similar at baseline on important characteristics	No	Yes	Yes	No
Overall drop-out rate at endpoint was ≤20% for treatment group	No	No	No	No
Differential drop-out rate between groups at endpoint was ≤15% or lower	Yes	Yes	No	No
Adherence to the intervention protocols were high	No	Yes	No	No
Other interventions were avoided or similar in the groups	Yes	Yes	Yes	Yes
Outcomes were assessed using valid and reliable measures	Yes	Yes	Yes	Yes
Sufficient sample size to be able to detect a difference with ≥80% power	Yes	No	Yes	Yes
Outcomes reported or subgroups analyzed were prespecified	Unclear	Unclear	Yes	Yes
All randomized participants were analyzed in the original group (intention-to-treat analysis)	Yes	Yes	Yes	Yes
Suggesting Level of Quality (% of criteria met)	Fair (62%)	Fair (62%)	Good (77%)	Poor (38%)
Level of Evidence	II	II	I	II

*Note.* Level of quality was defined as: Good Quality (75–100% criteria met), Fair (50–74% criteria met), Poor (25–49% criteria met).

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
