# Peer review of "Feasibility, Process, and Effects of Short-Term Calorie Reduction in Cancer Patients Receiving Chemotherapy: An Integrative Review"

_nutrients, 2020, doi:10.3390/nu12092823_

Round 1

Reviewer 1 Report

From my point of view the paper needs one important in the discussion. Fasting can lead to a lower tumor cell proliferation rate and a lower tumor proliferation rate may impair the efficacy of chemotherapy.

Author Response

Thank you so much for the feedback! Attached please see our response.

Reviewer 2 Report

This is an interesting study summarizes the results of studies on the Effects of Short-Term Calorie Reduction in Cancer Patients Receiving Chemotherapy. Please see my suggestions below

  1. Is there any information about nutritional supplements that the participants used or suggested to use during the fasting period or in general in the reported studies?
  2. Can you add some more information about other types of recommended diets for cancer since you have a limited number of references on SCR? 
  3. line 244, I suggest putting "several" instead of "many" since there are just four studies reported.
  4. line 265, Please add more details about the following statement " patients who had fasted for more than 24 hours showed a faster recovery" which studies/study showed that. The other study stated that this is just for the patients who had fasted for longer than 48 hours, what similarities or differences do these studies have?

Author Response

Thank you so much for your feedback and encouragement. These comments are thought provoking and we have revised the manuscript accordingly. Attached please find our point-be-point response.

Reviewer 3 Report

Materials and Methods

My congratulations to the authors, it was a pleasure to read in depth your complete and exhaustive work.

Methods

106 Table 1. Searched Databases, Searching Strategies, and the Number of Initial Results Results. Why the authors used the same search terms on all databases? For example, cancer term is a highlighted mesh in this study but was not included in all the databases.

100 reviewed. We did not set any limits on the dates of publication and the final date of search is on  August 6th, 2020

143 Among the seven studies published between 2009 and 2020

The two sentences above are confusing, please modify it.

Discussion

321 Most of the reviewed studies… Needs references

325 normal cells. A couple of studies…Needs references

Author Response

(The authors gave the same response as above.)

Round 2

Reviewer 2 Report

The authors did not address my second suggestion. 

2. Can you add some more information about other types of recommended diets for cancer since you have a limited number of references on SCR?

Author Response

Dear reviewer,

In the last revision of the manuscript, we have actually added relevant information according to this excellent recommendation (line 314-318) but failed to explain it in the point-by-point reply (we accidentally copied and pasted the reply to recommendation 1 twice). Our apologies for the confusion! Attached please see our detailed reply. Thank you!
